# Probabilistic Variational Bounds
# for Graphical Models

**Qiang Liu**
Computer Science
Dartmouth College
qliu@cs.dartmouth.edu

**John Fisher III**
CSAIL
MIT
fisher@csail.mit.edu

**Alexander Ihler**
Computer Science
Univ. of California, Irvine
ihler@ics.uci.edu

## Abstract

Variational algorithms such as tree-reweighted belief propagation can provide deterministic bounds on the partition function, but are often loose and difficult to use in an "any-time" fashion, expending more computation for tighter bounds. On the other hand, Monte Carlo estimators such as importance sampling have excellent any-time behavior, but depend critically on the proposal distribution. We propose a simple Monte Carlo based inference method that augments convex variational bounds by adding importance sampling (IS). We argue that convex variational methods naturally provide good IS proposals that "cover" the target probability, and reinterpret the variational optimization as designing a proposal to minimize an upper bound on the variance of our IS estimator. This both provides an accurate estimator and enables construction of any-time probabilistic bounds that improve quickly and directly on state-of-the-art variational bounds, and provide certificates of accuracy given enough samples relative to the error in the initial bound.

## 1   Introduction

Graphical models such as Bayesian networks, Markov random fields and deep generative models provide a powerful framework for reasoning about complex dependency structures over many variables [see e.g., 14, 13]. A fundamental task is to calculate the partition function, or normalization constant. This task is #P-complete in the worst case, but in many practical cases it is possible to find good deterministic or Monte Carlo approximations. The most useful approximations should give not only accurate estimates, but some form of confidence interval, so that for easy problems one has a certificate of accuracy, while harder problems are identified as such. Broadly speaking, approximations fall into two classes: variational optimization, and Monte Carlo sampling.

Variational inference [29] provides a spectrum of deterministic estimates and upper and lower bounds on the partition function; these include loopy belief propagation (BP), which is often quite accurate; its convex variants, such as tree reweighted BP (TRW-BP), which give upper bounds on the partition function; and mean field type methods that give lower bounds. Unfortunately, these methods often lack useful accuracy assessments; although in principle a pair of upper and lower bounds (such as TRW-BP and mean field) taken together give an interval containing the true solution, the gap is often too large to be practically useful. Also, improving these bounds typically means using larger regions, which quickly runs into memory constraints.

Monte Carlo methods, often based on some form of importance sampling (IS), can also be used to estimate the partition function [e.g., 15]. In principle, IS provides unbiased estimates, with the potential for a *probabilistic* bound: a bound which holds with some user-selected probability $1 - \delta$. Sampling estimates can also easily trade time for increased accuracy, without using more memory. Unfortunately, choosing the proposal distribution in IS is often both crucial and difficult; if poorly chosen, not only is the estimator high-variance, but the samples' empirical variance estimate is also misleading, resulting in both poor accuracy *and* poor confidence estimates; see e.g., [35, 1].

We propose a simple algorithm that combines the advantages of variational and Monte Carlo methods. Our result is based on an observation that convex variational methods, including TRW-BP and its generalizations, naturally provide good importance sampling proposals that "cover" the probability of the target distribution; the simplest example is a mixture of spanning trees constructed by TRW-BP. We show that the importance weights of this proposal are uniformly bounded by the convex upper bound itself, which admits a bound on the variance of the estimator, and more importantly, allows the use of exponential concentration inequalities such as the empirical Bernstein inequality to provide explicit confidence intervals. Our method provides several important advantages:

First, the upper bounds resulting from our sampling approach improve directly on the initial variational upper bound. This allows our bound to start at a state-of-the-art value, and be quickly and easily improved in an any-time, memory efficient way. Additionally, using a two-sided concentration bound provides a "certificate of accuracy" which improves over time at an easily analyzed rate. Our upper bound is significantly better than existing probabilistic upper bounds, while our corresponding lower bound is typically worse with few samples but eventually outperforms state-of-the-art probabilistic bounds [11].

Our approach also results in improved estimates of the partition function. As in previous work [32, 34, 31], applying importance sampling serves as a "bias correction" to variational approximations. Here, we interpret the variational bound optimization as equivalent to minimizing an upper bound on the IS estimator's variance. Empirically, this translates into estimates that can be significantly more accurate than IS using other variational proposals, such as mean field or belief propagation.

**Related Work.** Importance sampling and related approaches have been widely explored in the Bayesian network literature, in which the partition function corresponds to the probability of observed evidence; see e.g., [8, 26, 33, 11] and references therein. Dagum and Luby [4] derive a sample size to ensure a probabilistic bound with given relative accuracy; however, they use the normalized Bayes net distribution as a proposal, leading to prohibitively large numbers of samples when the partition function is small, and making it inapplicable to Markov random fields. Cheng [2] refines this result, including a user-specified bound on the importance weights, but leaves the choice of proposal unspecified.

Some connections between IS and variational methods are also explored in Yuan and Druzdzel [32, 34], Wexler and Geiger [31], Gogate and Dechter [11], in which proposals are constructed based on loopy BP or mean field methods. While straightforward in principle, we are not aware of any prior work which uses variational upper bounds to construct a proposal, or more importantly, analyzes their properties. An alternative probabilistic upper bound can be constructed using "perturb and MAP" methods [23, 12] combined with recent concentration results [22]; however, in our experiments the resulting bounds were quite loose. Although not directly related to our work, there are also methods that connect variational inference with MCMC [e.g., 25, 6].

Our work is orthogonal to the line of research on adaptive importance sampling, which refines the proposal as more samples are drawn [e.g., 21, 3]; we focus on developing a good fixed proposal based on variational ideas, and leave adaptive improvement as a possible future direction.

**Outline.** We introduce background on graphical models in Section 2. Our main result is presented in Section 3, where we construct a tree reweighted IS proposal, discuss its properties, and propose our probabilistic bounds based on it. We give a simple extension of our method to higher order cliques based on the weighted mini-bucket framework in Section 4. We then show experimental comparisons in Section 5 and conclude with Section 6.

## 2 Background

### 2.1 Undirected Probabilistic Graphical Models

Let $\boldsymbol{x} = [x_1, \ldots, x_p]$ be a discrete random vector taking values in $\mathcal{X} \overset{def}{=} \mathcal{X}_1 \times \cdots \times \mathcal{X}_p$; a probabilistic graphical model on $\boldsymbol{x}$, in an over-complete exponential family form, is

$$p(\boldsymbol{x}; \boldsymbol{\theta}) = \frac{f(\boldsymbol{x}; \boldsymbol{\theta})}{Z(\boldsymbol{\theta})}, \quad \text{with} \quad f(\boldsymbol{x}; \boldsymbol{\theta}) = \exp\Big( \sum_{\alpha \in \mathcal{I}} \theta_\alpha(\boldsymbol{x}_\alpha) \Big), \quad Z(\boldsymbol{\theta}) = \sum_{\boldsymbol{x} \in \mathcal{X}} f(\boldsymbol{x}; \boldsymbol{\theta}), \quad (1)$$

where $\mathcal{I} = \{\alpha\}$ is a set of subsets of variable indices, and $\theta_\alpha\colon \mathcal{X}_\alpha \to \mathbb{R}$ are functions of $\boldsymbol{x}_\alpha$; we denote by $\boldsymbol{\theta} = \{\theta_\alpha(\boldsymbol{x}_\alpha)\colon \forall \alpha \in \mathcal{I}, \boldsymbol{x}_\alpha \in \mathcal{X}_\alpha\}$ the vector formed by the elements of $\theta_\alpha(\cdot)$, called the *natural parameters*. Our goal is to calculate the partition function $Z(\boldsymbol{\theta})$ that normalizes the distribution; we often drop the dependence on $\boldsymbol{\theta}$ and write $p(\boldsymbol{x}) = f(\boldsymbol{x})/Z$ for convenience.

The factorization of $p(\boldsymbol{x};\boldsymbol{\theta})$ can be represented by an undirected graph $G = (V, E_G)$, called its *Markov graph*, where each vertex $k \in V$ is associated with a variable $x_k$, and nodes $k, l \in V$ are connected (i.e., $(kl) \in E_G$) iff there exists some $\alpha \in \mathcal{I}$ that contains both $k$ and $l$; then, $\mathcal{I}$ is a set of cliques of $G$. A simple special case of (1) is the pairwise model, in which $\mathcal{I} = V \cup E$:

$$f(\boldsymbol{x};\boldsymbol{\theta}) = \exp\big(\sum_{i \in V} \theta_k(x_k) \; + \sum_{(kl) \in E_G} \theta_{kl}(x_k, x_l)\big). \tag{2}$$

## 2.2 Monte Carlo Estimation via Importance Sampling

Importance sampling (IS) is at the core of many Monte Carlo methods for estimating the partition function. The idea is to take a tractable, normalized distribution $q(\boldsymbol{x})$, called the *proposal*, and estimate $Z$ using samples $\{\boldsymbol{x}^i\}_{i=1}^n \sim q(\boldsymbol{x})$:

$$\hat{Z} = \frac{1}{n}\sum_{i=1}^n w(\boldsymbol{x}^i), \qquad \text{with} \qquad w(\boldsymbol{x}^i) = \frac{f(\boldsymbol{x}^i)}{q(\boldsymbol{x}^i)},$$

where $w(\boldsymbol{x})$ is called the importance weight. It is easy to show that $\hat{Z}$ is an unbiased estimator of $Z$, in that $\mathbb{E}\hat{Z} = Z$, if $q(x) > 0$ whenever $p(x) > 0$, and has a MSE of $\mathbb{E}(\hat{Z} - Z)^2 = \frac{1}{n}\mathrm{var}(w(\boldsymbol{x}))$.

Unfortunately, the IS estimator often has very high variance if the choice of proposal distribution is very different from the target, especially when the proposal is more peaked or has thinner tails than the target. In these cases, there exist configurations $\boldsymbol{x}$ such that $q(\boldsymbol{x}) \ll p(\boldsymbol{x})$, giving importance weights $w(\boldsymbol{x}) = f(\boldsymbol{x})/q(\boldsymbol{x})$ with extremely large values, but very small probabilities. Due to the low probability of seeing these large weights, a "typical" run of IS often underestimates $Z$ in practice, that is, $\hat{Z} \leq Z$ with high probability, despite being unbiased.

Similarly, the empirical variance of $\{w(\boldsymbol{x}^i)\}$ can also severely underestimate the true variance $\mathrm{var}(w(\boldsymbol{x}))$, and so fail to capture the true uncertainty of the estimator. For this reason, concentration inequalities that make use of the empirical variance (see Section 3) also require that $w$, or its variance, be bounded. It is thus desirable to construct proposals that are similar to, and less peaked than, the target distribution $p(\boldsymbol{x})$. The key observation of this work is to show that tree reweighted BP and its generalizations provide a easy way to construct such good proposals.

## 2.3 Tree Reweighted Belief Propagation

Next we describe the tree reweighted (TRW) upper bound on the partition function, restricting to pairwise models (2) for notational ease. In Section 4 we give an extension that includes both more general factor graphs, and more general convex upper bounds.

Let $\mathcal{T} = \{T\}$ be a set of spanning trees $T = (V, E_T)$ of $G$ that covers $G$: $\cup_T E_T = E_G$. We assign a set of nonnegative weights $\{\rho^T : T \in \mathcal{T}\}$ on $\mathcal{T}$ such that $\sum_T \rho^T = 1$. Let $\boldsymbol{\theta}^{\mathcal{T}} = \{\boldsymbol{\theta}^T : T \in \mathcal{T}\}$ be a set of natural parameters that satisfies $\sum_T \rho^T \boldsymbol{\theta}^T = \boldsymbol{\theta}$, and each $\boldsymbol{\theta}^T$ respects the structure of $T$ (so that $\theta_{kl}^T(x_k, x_l) \equiv 0$ for $\forall(kl) \notin E_T$). Define

$$p^T(\boldsymbol{x}) \overset{def}{=} p(\boldsymbol{x};\boldsymbol{\theta}^T) = \frac{f(\boldsymbol{x};\boldsymbol{\theta}^T)}{Z(\boldsymbol{\theta}^T)}, \quad \text{with} \quad f(\boldsymbol{x};\boldsymbol{\theta}^T) = \exp\Big(\sum_{k \in V} \theta_k^T(x_k) + \sum_{(kl) \in E_T} \theta_{kl}^T(x_k, x_l)\Big);$$

then $p^T(\boldsymbol{x})$ is a tree structured graphical model with Markov graph $T$. Wainwright et al. [30] use the fact that $\log Z(\theta)$ is a convex function of $\theta$ to propose to upper bound $\log Z(\boldsymbol{\theta})$ by

$$\log Z_{trw}(\boldsymbol{\theta}^{\mathcal{T}}) = \sum_{T \in \mathcal{T}} \rho^T \log Z(\boldsymbol{\theta}^T) \quad \geq \quad \log Z(\sum_{T \in \mathcal{T}} \rho^T \boldsymbol{\theta}^T) = \log Z(\boldsymbol{\theta}),$$

via Jensen's inequality. Wainwright et al. [30] find the tightest bound via a convex optimization:

$$\log Z^*_{trw}(\boldsymbol{\theta}) = \min_{\boldsymbol{\theta}^{\mathcal{T}}} \left\{ \log Z_{trw}(\boldsymbol{\theta}^{\mathcal{T}}), \quad s.t. \quad \sum_T \rho^T \boldsymbol{\theta}^T = \boldsymbol{\theta} \right\}. \tag{3}$$

Wainwright et al. [30] solve this optimization by a tree reweighted belief propagation (TRW-BP) algorithm, and note that the optimality condition of (3) is equivalent to enforcing a marginal consistency condition on the trees – a $\boldsymbol{\theta}^{\mathcal{T}}$ optimizes (3) if and only if there exists a set of common singleton and pairwise "pseudo-marginals" $\{b_k(x_k), \ b_{kl}(x_k, x_l)\}$, corresponding to the fixed point of TRW-BP in Wainwright et al. [30], such that

$$b(x_k, x_l) = p^T(x_k, x_l), \quad \forall (kl) \in T, \qquad \text{and} \qquad b(x_k) = p^T(x_k), \quad \forall k \in V,$$

where $p^T(x_k)$ and $p^T(x_k, x_l)$ are the marginals of $p^T(\boldsymbol{x})$. Thus, after running TRW-BP, we can calculate $p^T(\boldsymbol{x})$ via

$$p^T(\boldsymbol{x}) = p(\boldsymbol{x} \ ; \ \boldsymbol{\theta}^T) = \prod_{k \in V} b_k(x_k) \prod_{kl \in E_T} \frac{b_{kl}(x_k, x_l)}{b_k(x_k) b_l(x_l)}. \tag{4}$$

Because TRW provides a convex upper bound, it is often well-suited to the inner loop of learning algorithms [e.g., 28]. However, it is often far less accurate than its non-convex counterpart, loopy BP; in some sense, this can be viewed as the cost of being a bound. In the next section, we show that our importance sampling procedure can "de-bias" the TRW bound, to produce an estimator that significantly outperforms loopy BP; in addition, due to the nice properties of our TRW-based proposal, we can use an empirical Bernstein inequality to construct a non-asymptotic confidence interval for our estimator, turning the deterministic TRW bound into a much tighter probabilistic bound.

## 3 Tree Reweighted Importance Sampling

We propose to use the collection of trees $p^T(\boldsymbol{x})$ and weights $\rho^T$ in TRW to form an importance sampling proposal,

$$q(\boldsymbol{x}; \boldsymbol{\theta}^{\mathcal{T}}) = \sum_{T \in \mathcal{T}} \rho^T p^T(\boldsymbol{x}), \tag{5}$$

which defines an estimator $\hat{Z} = \frac{1}{n} \sum_{i=1}^n w(\boldsymbol{x}^i)$ with $\boldsymbol{x}^i$ drawn i.i.d. from $q(\boldsymbol{x}; \boldsymbol{\theta}^{\mathcal{T}})$. Our observation is that this proposal is *good* due to the special convex construction of TRW. To see this, we note that the reparameterization constraint $\sum_T \rho^T \boldsymbol{\theta}^T = \boldsymbol{\theta}$ can be rewritten as

$$f(\boldsymbol{x}; \boldsymbol{\theta}) = Z_{trw}(\boldsymbol{\theta}^{\mathcal{T}}) \prod_T \left[ p^T(\boldsymbol{x}) \right]^{\rho^T}, \tag{6}$$

that is, $f(\boldsymbol{x}; \boldsymbol{\theta})$ is the $\{\rho^T\}$-weighted geometric mean of $p^T(\boldsymbol{x})$ up to a constant $Z_{trw}$; on the other hand, $q(\boldsymbol{x}; \boldsymbol{\theta}^{\mathcal{T}})$, by its definition, is the arithmetic mean of $p^T(\boldsymbol{x})$, and hence will always be larger than the geometric mean by the AM-GM inequality, guaranteeing good coverage of the target's probability. To be specific, we have $q(\boldsymbol{x}; \boldsymbol{\theta}^{\mathcal{T}})$ is always no smaller than $f(\boldsymbol{x}; \boldsymbol{\theta})/Z_{trw}(\boldsymbol{\theta}^{\mathcal{T}})$, and hence the importance weight $w(\boldsymbol{x})$ is always upper bounded by $Z_{trw}(\boldsymbol{\theta}^{\mathcal{T}})$. Note that (5)–(6) immediately implies that $q(x; \boldsymbol{\theta}^{\mathcal{T}}) > 0$ whenever $f(\boldsymbol{x}; \boldsymbol{\theta}) > 0$. We summarize our result as follows.

**Proposition 3.1.** *i) If $\sum_T \rho^T \boldsymbol{\theta}^T = \boldsymbol{\theta}$, $\rho^T \geq 0$, $\sum_T \rho^T = 1$, then the importance weight $w(\boldsymbol{x}) = f(\boldsymbol{x}; \boldsymbol{\theta})/q(\boldsymbol{x}; \boldsymbol{\theta}^{\mathcal{T}})$, with $q(\boldsymbol{x}; \boldsymbol{\theta}^{\mathcal{T}})$ defined in (5), satisfies*

$$w(\boldsymbol{x}) \leq Z_{trw}(\boldsymbol{\theta}^{\mathcal{T}}), \quad \forall \boldsymbol{x} \in \mathcal{X}, \tag{7}$$

*that is, the importance weights of (5) are always bounded by the TRW upper bound; this reinterprets the TRW optimization (3) as finding the mixture proposal in (5) that has the smallest upper bound on the importance weights.*

*ii) As a result, we have $\max\{\text{var}(w(\boldsymbol{x})), \widehat{\text{var}}(w(\boldsymbol{x}))\} \leq \frac{1}{4} Z^2_{trw}$ for $\boldsymbol{x} \sim q(x; \boldsymbol{\theta}^{\mathcal{T}})$, where $\widehat{\text{var}}(w(\boldsymbol{x}))$ is the empirical variance of the weights. This implies that $\mathbb{E}(\hat{Z} - Z)^2 \leq \frac{1}{4n} Z^2_{trw}$.*

*Proof.* i) Directly apply AM-GM inequality on (5) and (6). ii) Note that $\mathbb{E}(w(\boldsymbol{x})) = Z$ and hence $\mathrm{var}(w(\boldsymbol{x})) = \mathbb{E}(w(\boldsymbol{x})^2) - \mathbb{E}(w(\boldsymbol{x}))^2 \le Z_{trw}Z - Z^2 \le \frac{1}{4}Z_{trw}^2$.  □

Note that the TRW reparameterization (6) is key to establishing our results. Its advantage is two-fold: First, it provides a simple upper bound on $w(\boldsymbol{x})$; for an arbitrary $q(\cdot)$, establishing such an upper bound may require a difficult combinatorial optimization over $\boldsymbol{x}$. Second, it enables that bound to be optimized over $q(\cdot)$, resulting in a good proposal.

**Empirical Bernstein Confidence Bound.**  The upper bound of $w(\boldsymbol{x})$ in Proposition 3.1 allows us to use exponential concentration inequalities and construct tight finite-sample confidence bounds. Based on the empirical Bernstein inequality in Maurer and Pontil [19], we have

**Corollary 3.2** (Maurer and Pontil [19]). *Let $\hat{Z}$ be the IS estimator resulting from $q(\boldsymbol{x})$ in (5). Define*

$$\Delta = \sqrt{\frac{2\widehat{\mathrm{var}}(w(\boldsymbol{x}))\log(2/\delta)}{n}} + \frac{7Z_{trw}(\boldsymbol{\theta}^{\mathcal{T}})\log(2/\delta)}{3(n-1)}, \tag{8}$$

*where $\widehat{\mathrm{var}}(w(\boldsymbol{x})$ is the empirical variance of the weights, then $\hat{Z}_+ = \hat{Z} + \Delta$ and $Z_- = \hat{Z} - \Delta$ are upper and lower bounds of $Z$ with at least probability $(1 - \delta)$, respectively, that is, $\Pr(Z \le \hat{Z}_+) \ge 1 - \delta$ and $\Pr(\hat{Z}_- \le Z) \ge 1 - \delta$.*

The quantity $\Delta$ is quite intuitive, with the first term proportional to the empirical standard deviation and decaying at the classic $1/\sqrt{n}$ rate. The second term captures the possibility that the empirical variance is inaccurate; it depends on the boundedness of $w(\boldsymbol{x})$ and decays at rate $1/n$. Since $\widehat{\mathrm{var}}(w) < Z_{trw}^2$, the second term typically dominates for small $n$, and the first term for large $n$.

When $\Delta$ is large, the lower bound $\hat{Z} - \Delta$ may be negative; this is most common when $n$ is small and $Z_{trw}$ is much larger than $Z$. In this case, we may replace $\hat{Z}_-$ with any deterministic lower bound, or with $\hat{Z}\delta$, which is a $(1 - \delta)$ probabilistic bound by the Markov inequality; see Gogate and Dechter [11] for more Markov inequality based lower bounds. However, once $n$ is large enough, we expect $\hat{Z}_-$ should be much tighter than using Markov's inequality, since $\hat{Z}_-$ also leverages boundedness and variance information.[1] On the other hand, the Bernstein upper bound $\hat{Z}_+$ readily gives a good upper bound, and is usually much tighter than $Z_{trw}$ even with a relatively small $n$.

For example, if $\hat{Z} \ll Z_{trw}$ (e.g., the TRW bound is not tight), our upper bound $\hat{Z}_+$ improves rapidly on $Z_{trw}$ at rate $1/n$ and passes $Z_{trw}$ when $n \ge \frac{7}{3}\log(2/\delta) + 1$ (for example, for $\delta = 0.025$ used in our experiments, we have $\hat{Z}_+ \le Z_{trw}$ by $n = 12$). Meanwhile, one can show that the lower bound must be non-trivial ($\hat{Z}_- > 0$) if $n > 6(Z_{trw}/\hat{Z})\log(2/\delta) + 1$. During sampling, we can roughly estimate the point at which it will become non-trivial, by finding $n$ such that $\hat{Z} \ge \Delta$. More rigorously, one can apply a stopping criterion [e.g., 5, 20] on $n$ to guarantee a relative error $\epsilon$ with probability at least $1 - \delta$, using the bound on $w(\boldsymbol{x})$; roughly, the expected number of samples will depend on $Z_{trw}/Z$, the relative accuracy of the variational bound.

## 4  Weighted Mini-bucket Importance Sampling

We have so far presented our results for tree reweighted BP on pairwise models, which approximates the model using combinations of trees. In this section, we give an extension of our results to general higher order models, and approximations based on combinations of low-treewidth graphs. Our extension is based on the weighted mini-bucket framework [7, 17, 16], but extensions based on other higher order generalizations of TRW, such as Globerson and Jaakkola [9], are also possible. We only sketch the main idea in this section.

We start by rewriting the distribution using the chain rule along some order $o = [x_1, \dots, x_p]$,

$$f(\boldsymbol{x}) = Z \prod_k p(x_k | \boldsymbol{x}_{\mathrm{pa}(k)}). \tag{9}$$

where $\mathrm{pa}(k)$, called the induced parent set of $k$, is the set of variables adjacent to $x_k$ when it is eliminated along order $o$. The largest parent size $\omega := \max_{k \in V} |\mathrm{pa}(k)|$ is called the *induced width* of $G$ along order $o$, and the computational complexity of exact variable elimination along order $o$ is $O(\exp(\omega))$, which is intractable when $\omega$ is large.

Weighted mini-bucket is an approximation method that avoids the $O(\exp(\omega))$ complexity by splitting each $\mathrm{pa}(k)$ into several smaller "mini-buckets" $\overline{\mathrm{pa}}_\ell(k)$, such that $\cup_\ell \overline{\mathrm{pa}}_\ell(k) = \mathrm{pa}(k)$, where the size of the $\overline{\mathrm{pa}}_\ell(k)$ is controlled by a predefined number $ibound \geq |\overline{\mathrm{pa}}_\ell(k)|$, so that the $ibound$ trades off the computational complexity with approximation quality. We associate each $\overline{\mathrm{pa}}_\ell(k)$ with a nonnegative weight $\rho_{k\ell}$, such that $\sum_\ell \rho_{k\ell} = 1$. The weighted mini-bucket algorithm in Liu [16] then frames a convex optimization to output an upper bound $Z_{wmb} \geq Z$ together with a set of "pseudo-" conditional distributions $b_{k\ell}(x_k | \boldsymbol{x}_{\overline{\mathrm{pa}}_\ell(k)})$, such that

$$f(\boldsymbol{x}) = Z_{wmb} \prod_k \prod_\ell b_{k\ell}(x_k | \boldsymbol{x}_{\overline{\mathrm{pa}}_\ell(k)})^{\rho_{k\ell}}, \tag{10}$$

which, intuitively speaking, can be treated as approximating each conditional distribution $p(x_k | \boldsymbol{x}_{\mathrm{pa}(k)})$ with a geometric mean of the $b_{k\ell}(x_k | x_{\overline{\mathrm{pa}}_\ell(k)})$; while we omit the details of weighted mini-bucket [17, 16] for space, what is most important for our purpose is the representation (10).

Similarly to with TRW, we define a proposal distribution by replacing the geometric mean with an arithmetic mean:

$$q(\boldsymbol{x}) = \prod_k \sum_\ell \rho_{k\ell} \, b_{k\ell}(x_k | \boldsymbol{x}_{\overline{\mathrm{pa}}_\ell(k)}). \tag{11}$$

We can again use the AM-GM inequality to obtain a bound on $w(\boldsymbol{x})$, that $w(\boldsymbol{x}) \leq Z_{wmb}$.

**Proposition 4.1.** *Let* $w(\boldsymbol{x}) = f(\boldsymbol{x})/q(\boldsymbol{x})$*, where* $f(\boldsymbol{x})$ *and* $q(\boldsymbol{x})$ *satisfy* (10) *and* (11)*, with* $\sum_\ell \rho_{k\ell} = 1$*,* $\rho_{k\ell} \geq 0$*,* $\forall k, \ell$*. Then,*

$$w(\boldsymbol{x}) \leq Z_{wmb}, \quad \forall \boldsymbol{x} \in \mathcal{X}.$$

*Proof.* Use the AM-GM inequality, $\prod_\ell b_{k\ell}(x_k | \boldsymbol{x}_{\overline{\mathrm{pa}}_\ell(k)})^{\rho_{k\ell}} \leq \sum_\ell \rho_{k\ell} \, b_{k\ell}(x_k | \boldsymbol{x}_{\overline{\mathrm{pa}}_\ell(k)})$, for each $k$.

Note that the form of $q(\boldsymbol{x})$ makes it convenient to sample by sequentially drawing each variable $x_k$ from the mixture $\sum_\ell \rho_{k\ell} \, b_{k\ell}(x_k | \boldsymbol{x}_{\overline{\mathrm{pa}}_\ell(k)})$ along the reverse order $[x^p, \ldots, x^1]$. The proposal $q(\boldsymbol{x})$ also can be viewed as a mixture of a large number of models with induced width controlled by $ibound$; this can be seen by expanding the form in (11),

$$q(\boldsymbol{x}) = \sum_{\ell_1 \cdots \ell_p} \rho_{\ell_1 \cdots \ell_p} q_{\ell_1 \cdots \ell_p}(\boldsymbol{x}), \quad \text{where} \quad \rho_{\ell_1 \cdots \ell_p} = \prod_k \rho_{k\ell_k}, \quad q_{\ell_1 \cdots \ell_p}(\boldsymbol{x}) = \prod_k b_{k\ell_k}(x_k | x_{\overline{\mathrm{pa}}_\ell(k)}).$$

## 5 Experiments

We demonstrate our algorithm using synthetic Ising models, and real-world models from recent UAI inference challenges. We show that our TRW proposal can provide better estimates than other proposals constructed from mean field or loopy BP, particularly when it underestimates the partition function; in this case, the proposal may be too peaked and fail to approach the true value even for extremely large sample sizes $n$. Using the empirical Bernstein inequality, our TRW proposal also provides strong probabilistic upper and lower bounds. When the model is relatively easy or $n$ is large, our upper and lower bounds are close, demonstrating the estimate has high confidence.

### 5.1 MRFs on $10 \times 10$ Grids

We illustrate our method using pairwise Markov random fields (2) on a $10 \times 10$ grid. We start with a simple Ising model with $\theta_k(x_k) = \sigma_s x_k$ and $\theta_{kl}(x_k, x_l) = \sigma_p x_k x_l$, $x_k \in \{-1, 1\}$, where $\sigma_s$ represents the external field and $\sigma_p$ the correlation. We fix $\sigma_s = 0.01$ and vary $\sigma_p$ from $-1.5$ (strong negative correlation) to $1.5$ (strong positive correlation). Different $\sigma_p$ lead to different inference hardness: inference is easy when the correlation is either very strong ($|\sigma_p|$ large) or very weak ($|\sigma_p|$ small), but difficult for an intermediate range of values, corresponding to a phase transition.

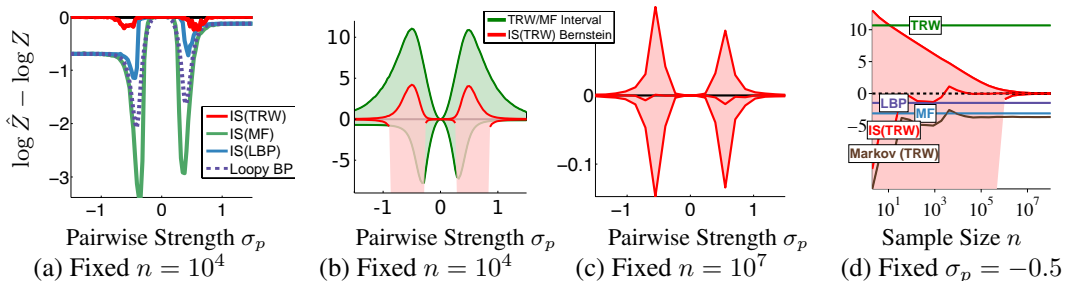

(a) Fixed $n = 10^4$     (b) Fixed $n = 10^4$     (c) Fixed $n = 10^7$     (d) Fixed $\sigma_p = -0.5$

Figure 1: Experiments on $10 \times 10$ Ising models with interaction strength $\sigma_p$ ranging from strong negative (-1.5) to strong positive (1.5).

We first run the standard variational algorithms, including loopy BP (LBP), tree reweighted BP (TRW), and mean field (MF). We then calculate importance sampling estimators based on each of the three algorithms. The TRW trees are chosen by adding random spanning trees until their union covers the grid; we assign uniform probability $\rho^T$ to each tree. The LBP proposal follows Gogate [10], constructing a (randomly selected) tree structured proposal based on the LBP pseudo-marginals. The MF proposal is $q(\boldsymbol{x}) = \prod_{k \in V} q_k(x_k)$, where the $q_k(x_k)$ are the mean field beliefs.

Figure 1(a) shows the result of the IS estimates based on a relatively small number of importance samples ($n = 10^4$). In this case the TRW proposal outperforms both the MF and LBP proposals; all the methods degrade when $\sigma_p \approx \pm.5$, corresponding to inherently more difficult inference. However, the TRW proposal converges to the correct values when the correlation is strong (e.g., $|\sigma_p| > 1$), while the MF and LBP proposals underestimate the true value, indicating that the MF and LBP proposals are too peaked, and miss a significant amount of probability mass of the target.

Examining the deterministic estimates, we note that the LBP approximation, which can be shown to be a lower bound on these models [27, 24], is also significantly worse than IS with the TRW proposal, and slightly worse than IS based on the LBP proposal. The TRW and MF bounds, of course, are far less accurate compared to either LBP or the IS methods, and are shown separately in Figure 1(b). This suggests it is often beneficial to follow the variational procedure with an importance sampling process, and use the corresponding IS estimators instead of the variational approximations to estimate the partition function.

Figure 1(b) compares the $95\%$ confidence interval of the IS based on the TRW proposal (filled with red), with the interval formed by the TRW upper bound and the MF lower bound (filled with green). We can see that the Bernstein upper bound is much tighter than the TRW upper bound, although at the cost of turning a deterministic bound into a $(1 - \delta)$ probabilistic bound. On the other hand, the Bernstein interval fails to report a meaningful lower bound when the model is difficult ($\sigma_p \approx \pm 0.5$), because $n = 10^4$ is small relative to the difficulty of the model. As shown in Figure 1(c), our method eventually produces both tight upper and lower bounds as sample size increases.

Figure 1(d) shows the Bernstein bound as we increase $n$ on a fixed model with $\sigma_p = -0.5$, which is relatively difficult according to Figure 1. Of the methods, our IS estimator becomes the most accurate by around $n = 10^3$ samples. We also show the Markov lower bound $\hat{Z}_{markov} = \hat{Z}\delta$ as suggested by Gogate [10]; it provides non-negative lower bounds for all sample sizes, but does not converge to the true value even with $n \to +\infty$ (in fact, it converges to $Z\delta$).

In addition to the simple Ising model, we also tested grid models with normally distributed parameters: $\theta_k(x_k) \sim \mathcal{N}(0, \sigma_s^2)$ and $\theta_{kl}(x_k, x_l) \sim \mathcal{N}(0, \sigma_p^2)$. Figure 2 shows the results when $\sigma_s = 0.01$ and we vary $\sigma_p$. In this case, LBP tends to overestimate the partition function, and IS with the LBP proposal performs quite well (similarly to our TRW IS); but with the previous example, this illustrates that it is hard to know whether BP will result in a high- or low-variance proposal. On this model, mean field IS is significantly worse and is not shown in the figure.

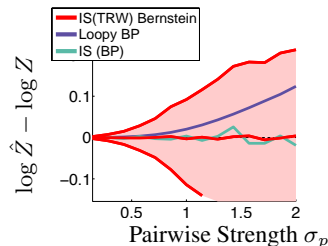

Figure 2: MRF with mixed interactions.

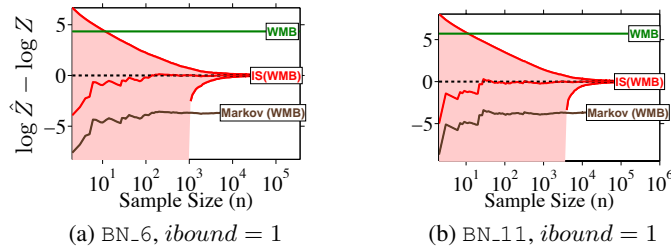

(a) BN_6, $ibound = 1$          (b) BN_11, $ibound = 1$

Figure 3: The Bernstein interval on (a) BN_6 and (b) BN_11 using $ibound = 1$ and different sample sizes $n$. These problems are relatively easy for variational approximations; we illustrate that our method gives tight bounds despite using no more memory than the original model.

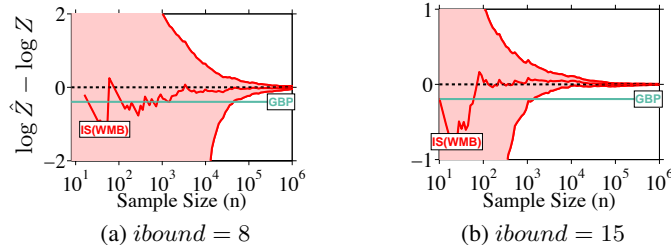

(a) $ibound = 8$          (b) $ibound = 15$

Figure 4: Results on a harder instance, pedigree20, at $ibound = 8, 15$ and different $n$.

## 5.2 UAI Instances

We test the weighted mini-bucket (WMB) version of our algorithm on instances from past UAI approximate inference challenges. For space reasons, we only report a few instances for illustration.

**BN Instances.** Figure 3 shows two Bayes net instances, BN_6 (true $\log Z = -58.41$) and BN_11 (true $\log Z = -39.37$). These examples are very easy for loopy BP, which estimates $\log Z$ nearly exactly, but of course gives no accuracy guarantees. For comparison, we run our WMB IS estimator using $ibound = 1$, e.g., cliques equal to the original factors. We find that we get tight confidence intervals by around $10^4$–$10^5$ samples. For comparison, the method of Dagum and Luby [4], using the normalized distribution as a proposal, would require samples proportional to $1/Z$: approximately $10^{25}$ and $10^{17}$, respectively.

**Pedigree Instances.** We next show results for our method on pedigree20, ($\log Z = -68.22$, induced width $\omega = 21$). and various $ibound$s; Figure 4 shows the results for $ibound$ 8 and 15. For comparision, we also evaluate GBP, defined on a junction graph with cliques found in the same way as WMB [18], and complexity controlled by the same $ibound$. Again, LBP and GBP generally give accurate estimates; the absolute error of LBP (not shown) is about 0.7, reducing to 0.4 and 0.2 at $ibound = 8$ and 15, respectively. The initial WMB bounds overestimate by 6.3 and 2.4 at $ibound = 8$ and 15, and are much less accurate. However, our method surpasses GBP's accuracy with a modest number of samples: for example, with $ibound = 15$ (Figure 4b), our IS estimator is more accurate than GBP with fewer than 100 samples, and our 95% Bernstein confidence interval passes GBP at roughly 1000 samples.

## 6 Conclusion

We propose a simple approximate inference method that augments convex variational bounds by adding importance sampling. Our formulation allows us to frame the variational optimization as designing a proposal that minimizes an upper bound on our estimator's variance, providing guarantees on the goodness of the resulting proposal. More importantly, this enables the construction of anytime probabilistic bounds that improve quickly and directly on state-of-the-art variational bounds, and provide certificates of accuracy given enough samples, relative to the error in the initial bound. One potential future direction is whether one can adaptively improve the proposal during sampling.

**Acknowledgement** This work is supported in part by VITALITE, under the ARO MURI program (Award number W911NF-11-1-0391); NSF grants IIS-1065618 and IIS-1254071; and by the United States Air Force under Contract No. FA8750-14-C-0011 under the DARPA PPAML program.

## Footnotes

[1] The Markov lower bounds by Gogate and Dechter [11] have the undesirable property that they may not become tighter with increasing $n$, and may even decrease.

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
