[Reviews · NeurIPS 2015]

Submitted by Assigned_Reviewer_1

The paper is very clearly written. There has been previous attempt at using the result of variational approximation to construct

proposal importance sampling distribution, but the importance of variational upperbound has not been discussed before. This seems

significant, as the theory and the empirical result demonstrates.

Further comments:

- when you say "any-time" algorithm, it is really an anytime improvement of the variational method (since one would

to run the variational method first).

- please report on running time. in particular, please report the time of running the variational method vs the time spent improving

then via IS (which will depend on the number of samples of course).

- all figures: please use different line styles for actual estimate and confidence bound. sometimes it is a bit hard to read

whether a line indicates a confidence bound or an actual estimate.

- under section 3, comment on how one would sample from q(). For example to do this efficiently would entail that a small number of trees are used.

Summary: The authors propose to use variational upperbound (e.g., TRW tree mixture) as the importance sampling proposal distribution.

They then demonstrated that this yields finite sample bound, works well in practice, and improves upon

usually loose upperbound approximation. It is the novel combination of two existing ideas that make this paper interesting.

Submitted by Assigned_Reviewer_2

The authors propose using an importance sampling procedure based on proposal distributions obtained by running TRW/BP to estimate the partition function of graphical models.

They demonstrate experimentally that such methods are effective at constructing good estimates of the partition function for easy instances but are also useful for more challenging instances (though at the price of higher sample complexity).

The paper is well-written and more or less self-contained.

General thoughts:

-There isn't much deep in the way of theoretical content here, so it would have been nice to see how IS with a TRW/BP proposal performs versus other common proposals that are used to estimate partition functions (there is quite a bit of work in this area).

The authors make a good point in their rebuttal that this approach does provide bounds which is arguably more useful than simple convergence rates.

-Having to actually run TRW/BP to convergence to get the proposal seems unfortunate, and in practice, these methods aren't necessarily particularly fast (especially for large models).

Once you do this, you need a large number of additional samples (if the model is hard) to get tight bounds.

So, I'm not sure how practical this approach really is.

-Z_TRW can be an exponentially bad bound (especially for very frustrated models).

This means that \Delta could be quite large.

Do you have some sense as to how the number of samples scales with the number of nodes and interaction strength on a single highly frustrated cycle?

-On page 7, you mention that LBP is a lower bound on these models and cite [26].

Sudderth et al. [26] showed only part of the claimed result under certain conditions on LBP fixed points.

The result you are claiming was actually proved later on in:

N. Ruozzi.

The Bethe partition function of log-supermodular graphical models.

Advances in Neural Information Processing Systems (NIPS), December 2012.

Typos:

-pg. 1, "optimization and Monte Carlo" -> "optimization and Monte Carlo". -pg. 1, "mean field type" -> "mean-field type". -pg. 2, "important advantages:" -> "important advantages". -pg. 3,

It is helpful for readers when expectations use a subscript to denote the distribution with which the expectation is being taken. -pg. 8, "state-of-the-art variational bounds," -> "state-of-the-art variational bounds"
Summary: The authors propose using an importance sampling procedure based on proposal distributions obtained by running TRW/BP to estimate the partition function of graphical models.

The approach seems interesting, but it remains unclear how practical the approach is.

Submitted by Assigned_Reviewer_3

The authors propose to use the TRW variational distribution as a proposal distribution for importance sampling to obtain samples from a graphical model. The authors exploit the TRW structure of a convex combination of distributions over spanning trees to upper-bound the importance sampling weights, and thereby to obtain an empirical Bernstein confidence bound, which they show is superior to existing bounds in the literature. Careful experiments are performed on both synthetic and real data, and demonstrate the advantages of this method along many dimensions.

The paper is very well written; both the prose and mathematics are clear and easy to follow. The idea is original and elegant, and the technical results are developed in such a way that their techniques seem applicable to analyzing related sampling algorithms. The analysis is a valuable contribution.

My main concern with this paper is that importance sampling is a straw man for larger problems. One main selling point of this paper is its superiority to other importance sampling schemes. However, this is not a realistic baseline for large problems, and the authors should at least compare to MCMC methods, as they are also asymptotically unbiased. This paper would be stronger with an experiment with a large scale models, such as computer vision problems, and try their technique with only a small number of samples and compare the accuracy/computational time tradeoff. Alternatively, it would be interesting to compare

using TRW proposal in a Metropolis-Hastings scheme would Gibbs or Swendsen-Wang sampling for these problems, as all these methods can handle larger problems than importance sampling can.
Summary: The authors present an interesting combine to combine the computational advantages of variational inference with the unbiasedness of Monte Carlo methods, and in particular cleverly exploit convexity to derive finite-sample confidence bounds. Experiments are carefully performed and support their conclusions. The authors should consider how to apply their methods to larger scale (thousands, millions of nodes) graphical models, where it may only be feasible to draw a small number of samples.

Author Feedback
Author rebuttal: We thank all both the "heavy" and "light" reviewers for their comments.

Reviewer_1:

We compared to both LBP and MF proposals, which are the most obvious and commonly used baselines; see references in the submission. Our discussion on the BN instances (lines 410-412) also suggests that the method of Dagum & Luby [4] would perform much worse. Also, there is a distinction to be made between sampling-based estimators (of which there are many) and bounds (of which there are few). Complicated proposal constructions may improve estimator quality, but it would likely prove difficult to obtain a computable probability bound. With some work, for example, one can obtain a bound from the LBP or MF proposals, but these bounds appear much worse than TRW's and not practically useful (omitted from the submission for space).

We do not need to run TRWBP to convergence -- optimizing the TRW bound in Eq (3) gives the tightest bound on the weights, but our method can be applied to any mixture of trees that satisfies the reparameterization condition (i.e., \sum_T \rho_T \theta_T = \theta). For example, even uniformly splitting the original parameter vector is valid; a similar statement holds for the WMB variant.

TRW could give exponentially bad bounds in the worst case -- this is unavoidable due to the hardness of the problem. But our method can give tight, certified bounds in easy cases. Hard cases could need a lot of samples, but our method gives a clear estimate of how many samples may be needed, which still provides valuable information about the hardness of the problem and possibilities for further improvement.

4. Thanks for pointing out Ruozzi 2012. We will cite it, and fix the typos.

Reviewer_2:

1. The instances in the UAI approximate inference competitions are quite realistic and challenging (the largest UAI instance we tested has ~1200 nodes and ~40 tree width). We are also interested in testing even larger models (millions of nodes), but it is difficult to run exact inference for comparison on such models.

On the other hand, the behavior of our algorithm relative to TRW is very predictable, and should perform similarly on larger models. In particular, our bound in Eq. (8) gives a clear and non-asymptotic picture on how quality depends on the number of samples; for example, we just need n = 12 samples to beat the deterministic TRW upper bound (see line 247-250 and the left parts of the Fig. 3 when the sample size is small).

2. As with AR1, please note the distinction between producing an estimate versus a bound. MCMC is only asymptotically correct, and so it is very difficult to produce non-asymptotic error bounds from its samples. Also, note that our goal is to estimate the partition function, not expected values. It is actually not trivial to estimate the partition function from MCMC (or even i.i.d.) samples; the most straightforward method would be the harmonic mean method, or Chib's method, which can often perform quite badly.

3. We are also interested in adapting our idea to estimate expectations, in which case comparison with MCMC would be important and the idea of using a TRW proposal in Metropolis-Hastings, or as a part of adaptive MCMC, looks very promising.

Reviewer 7:

Good point about any-time; actually, both the sampling and variational stages are "any-time" (TRW and WMB don't need to converge for our method, and can be terminated early), but the balance between the stages is under the control of the user.

We would like to add a timing plot subject to space constraints. For reference, in our implementation at ibound 15 on pedigree20 (Fig 4b), we spend about 60 seconds on WMB, and our sampling rate is about 60 seconds for 100k samples.

The complexity of sampling from mixtures of trees is actually almost independent with the number of trees, since you can first decide which tree it belongs to and then sample from that particular tree. On the other hand, the process of constructing a large number of trees using TRW optimization in eq(3) can be very slow, but it can be done efficiently, independent of the numbers of trees, using the variational form in Wainwright [29] that only relies on the edge appearance probability.

On a related note, it is even more efficient to use the weighted mini-bucket (WMB) proposal in eq(11), which encodes a large number of trees using only a few parameters. This form simplifies optimizing over the tree weights (if desired), and sampling from eq(11) remains linear with the number of variables, and independent of the ibound (the tree width of the mixture components).